# Solving Large Sequential Games with the Excessive Gap Technique

**Christian Kroer, Gabriele Farina, and Tuomas Sandholm**
Department of Computer Science
Carnegie Mellon University
Pittsburgh, PA 15213
`{ckroer,gfarina,sandholm}@cs.cmu.edu`

## Abstract

There has been tremendous recent progress on equilibrium-finding algorithms for zero-sum imperfect-information extensive-form games, but there has been a puzzling gap between theory and practice. *First-order methods* have significantly better theoretical convergence rates than any *counterfactual-regret minimization (CFR)* variant. Despite this, CFR variants have been favored in practice. Experiments with first-order methods have only been conducted on small- and medium-sized games because those methods are complicated to implement in this setting, and because CFR variants have been enhanced extensively for over a decade they perform well in practice. In this paper we show that a particular first-order method, a state-of-the-art variant of the *excessive gap technique*—instantiated with the *dilated entropy distance function*—can efficiently solve large real-world problems competitively with CFR and its variants. We show this on large endgames encountered by the *Libratus* poker AI, which recently beat top human poker specialist professionals at no-limit Texas hold'em. We show experimental results on our variant of the excessive gap technique as well as a prior version. We introduce a numerically friendly implementation of the smoothed best response computation associated with first-order methods for extensive-form game solving. We present, to our knowledge, the first GPU implementation of a first-order method for extensive-form games. We present comparisons of several excessive gap technique and CFR variants.

## 1 Introduction

Two-player zero-sum extensive-form games (EFGs) are a general representation that enables one to model a myriad of settings ranging from security to business to military to recreational. The *Nash equilibrium* solution concept [22] prescribes a sound notion of rational play for this setting. It is also robust in this class of game: if the opponent plays some other strategy than an equilibrium strategy, that can only help us.

There has been tremendous recent progress on equilibrium-finding algorithms for extensive-form zero-sum games. However, there has been a vexing gap between the theory and practice of equilibrium-finding algorithms. In this paper we will help close that gap.

It is well-known that the strategy spaces of an extensive-form game can be transformed into convex polytopes that allow a bilinear saddle-point formulation (BSPP) of the Nash equilibrium problem as follows [26, 28, 16].

$$\min_{x \in \mathcal{X}} \max_{y \in \mathcal{Y}} \langle x, Ay \rangle = \max_{y \in \mathcal{Y}} \min_{x \in \mathcal{X}} \langle x, Ay \rangle \tag{1}$$

Problem (1) can be solved in a number of ways. Early on, von Stengel [28] showed that it can be solved with a linear program (LP)—by taking the dual of the optimization problem faced by one player (say the $y$ player) when holding the strategy of the $x$ player fixed, and injecting the primal $x$-player constraints into the dual LP. This approach was used in early work on extensive-form game solving, up to games of size $10^5$ [15]. Gilpin and Sandholm [10] coupled it with lossless abstraction in order to solve Rhode Island hold'em which has $10^9$ nodes in the game tree. Since then, LP approaches have fallen out of favor. The LP is often too large to fit in memory, and even when it does fit the iterations of the simplex or interior-point methods used to solve the LP take too long—even if only modest accuracy is required.

Instead, modern work on solving this game class in the large focuses on iterative methods that converge to a Nash equilibrium in the limit. Two types of algorithms have been popular in particular: regret-minimization algorithms based on *counterfactual regret minimization (CFR)* [29, 20, 1, 5, 21, 4], and *first-order methods (FOMs)* based on combining a fast *bilinear saddle-point problem (BSPP)* solver such as the *excessive gap technique (EGT)* [24] with an appropriate *distance-generating function (DGF)* for EFG strategies [11, 17, 19, 18].

The CFR family has been most popular in practice so far. The CFR$^+$ variant [27] was used to near-optimally solve heads-up limit Texas hold'em [1], a game that has $10^{13}$ decision points after lossless abstraction. CFR$^+$ was also used for subgame solving in Libratus [4], an AI that beat four top professional heads-up no-limit Texas hold'em poker (HUNL) players—a game that has $10^{161}$ decision points (before abstraction) [12]. A variant of CFR was also used by Libratus to compute the whole-game strategy (also known as "blueprint" strategy) [4]. A hybrid of CFR and CFR$^+$ was utilized by DeepStack to beat HUNL professionals [21].

CFR-based algorithms converge at a rate of $\frac{1}{\sqrt{T}}$, whereas some algorithms based on FOMs converge at a rate of $\frac{1}{T}$. Despite this theoretically superior convergence rate, FOMs have had relatively little adoption in practice. Comparisons of CFR-based algorithms and FOMs were conducted by Kroer et al. [17] and Kroer et al. [19], where they found that a heuristic variant of EGT instantiated with an appropriate distance measure is superior to CFR regret matching (RM) and CFR with regret-matching$^+$ (RM$^+$) for small-to-medium-sized games.

In this paper, we present the first experiments on a large game—a real game played by humans—showing that an aggressive variant of EGT instantiated with the DGF of Kroer et al. [19] is competitive with the CFR family in practice. It outperforms CFR with RM$^+$, although CFR$^+$ is still slightly faster. This is the first time that a FOM has been shown superior to any CFR variant on a real-world problem. We show this on subgames encountered by *Libratus*. The *Libratus* agent solved an abstraction of the full game of no-limit Texas hold'em ahead of time in order to obtain a "blueprint" strategy. During play, *Libratus* then refined this blueprint strategy by solving subgames with significantly more detailed abstractions in real time [4, 3]. Our experiments are on solving endgames encountered by *Libratus* in the beginning of the fourth ("river" in poker lingo) betting round, with the full fine-grained abstraction actually used by *Libratus*. This abstraction has no abstraction of cards, that is, the model captures all aspects of the cards. There is abstraction of bet sizes to keep the branching factor reasonable; in our experiments we use the exact full fine-grained betting abstraction that was used by *Libratus*. Thus we show that it is possible to get the theoretically superior guarantee of FOMs while also getting strong practical performance.

In order to make our approach practical, we introduce a number of practical techniques for running FOMs on EFGs. In particular, we derive efficient and numerically friendly expressions for the *smoothed-best response (SBR)* and *prox mapping*, two optimization subproblems that EGT solves at every iteration. Furthermore, we introduce a GPU-based variant of these operations which allows us to parallelize EGT iterations.

We show experiments for several variants of both EGT and CFR. For EGT, we consider two practical variants, one that has the initial smoothing parameter set optimistically, and one that additionally performs aggressive stepsizing. For CFR, we show experimental results for CFR with RM, RM$^+$, and CFR$^+$ (i.e., CFR with linear averaging and RM$^+$). We will describe these variants in detail in the body of the paper. We conducted all the experiments on parallelized GPU code.

## 2 Bilinear Saddle-Point Problems

The computation of a Nash equilibrium in a zero-sum imperfect-information EFG can be formulated as the following bilinear saddle-point problem:

$$\min_{x \in \mathcal{X}} \max_{y \in \mathcal{Y}} \langle x, Ay \rangle = \max_{y \in \mathcal{Y}} \min_{x \in \mathcal{X}} \langle x, Ay \rangle, \tag{2}$$

where $\mathcal{X}, \mathcal{Y}$ are convex, compact sets in Euclidean spaces $E_x, E_y$. $A$ is the sequence-form payoff matrix and $\mathcal{X}, \mathcal{Y}$ are the sequence-form strategy spaces of Player 1 and 2, respectively.

Several FOMs with attractive convergence properties have been introduced for BSPPs [25, 24, 23, 8]. These methods rely on having some appropriate distance measure over $\mathcal{X}$ and $\mathcal{Y}$, called a *distance-generating function* (DGF). Generally, FOMs use the DGF to choose steps: given a gradient and a scalar stepsize, a FOM moves in the negative gradient direction by finding the point that minimizes the sum of the gradient and of the DGF evaluated at the new point. In other words, the next step can be found by solving a regularized optimization problem, where long gradient steps are discouraged by the DGF. For EGT on EFGs, the DGF can be interpreted as a smoothing function applied to the best-response problems faced by the players.

**Definition 1.** *A distance-generating function for $\mathcal{X}$ is a function $d(x) : \mathcal{X} \to \mathbb{R}$ which is convex and continuous on $\mathcal{X}$, admits continuous selection of subgradients on the set $\mathcal{X}^\circ = \{x \in \mathcal{X} : \partial d(x) \neq \emptyset\}$, and has strong convexity modulus $\varphi$ w.r.t. $\|\cdot\|$. Distance-generating functions for $\mathcal{Y}$ are defined analogously.*

Given DGFs $d_{\mathcal{X}}, d_{\mathcal{Y}}$ for $\mathcal{X}, \mathcal{Y}$ with strong convexity moduli $\varphi_{\mathcal{X}}$ and $\varphi_{\mathcal{Y}}$ respectively, we now describe EGT [24] applied to (1). EGT forms two smoothed functions using the DGFs

$$f_{\mu_y}(x) = \max_{y \in \mathcal{Y}} \langle x, Ay \rangle - \mu_y d_{\mathcal{Y}}, \qquad \phi_{\mu_x}(y) = \min_{x \in \mathcal{X}} \langle x, Ay \rangle + \mu_x d_{\mathcal{X}}. \tag{3}$$

These functions are smoothed approximations to the optimization problem faced by the $x$ and $y$ player, respectively. The scalars $\mu_x, \mu_y > 0$ are smoothness parameters denoting the amount of smoothing applied. Let $y_{\mu_y}(x)$ and $x_{\mu_x}(y)$ refer to the $y$ and $x$ values attaining the optima in (3). These can be thought of as *smoothed best responses*. Nesterov [25] shows that the gradients of the functions $f_{\mu_y}(x)$ and $\phi_{\mu_x}(y)$ exist and are Lipschitz continuous. The gradient operators and Lipschitz constants are

$$\nabla f_{\mu_y}(x) = a_1 + A y_{\mu_y}(x), \qquad\qquad \nabla \phi_{\mu_x}(y) = a_2 + A^\top x_{\mu_x}(y),$$

$$L_1\left(f_{\mu_y}\right) = \frac{\|A\|^2}{\varphi_{\mathcal{Y}} \mu_y}, \qquad\qquad L_2\left(\phi_{\mu_x}\right) = \frac{\|A\|^2}{\varphi_{\mathcal{X}} \mu_x},$$

where $\|A\|$ is the $\ell_1$-norm operator norm.

Let the convex conjugate of $d_{\mathcal{X}} : \mathcal{X} \to \mathbb{R}$ be denoted by $d_{\mathcal{X}}^*(g) = \max_{x \in \mathcal{X}} g^T x - d(x)$. The gradient $\nabla d^*(g)$ of the conjugate then gives the solution to the smoothed-best-response problem.

Based on this setup, EGT minimizes the following saddle-point residual, which is equal to the sum of regrets for the players.

$$\epsilon_{\text{sad}}(x^t, y^t) = \max_{y \in \mathcal{Y}} (x^t)^T A y - \min_{x \in \mathcal{X}} x^T A y^t$$

The idea behind EGT is to maintain the *excessive gap condition* (EGC), $\text{EGV}(x, y) := \phi_{\mu_x}(y) - f_{\mu_y}(x) > 0$. The EGC implies a bound on the saddle-point residual: $\epsilon_{\text{sad}}(x^t, y^t) \leq \mu_x \Omega_{\mathcal{X}} + \mu_y \Omega_{\mathcal{Y}}$, where $\Omega_{\mathcal{X}} = \max_{x, x'} d_{\mathcal{X}}(x) - d_{\mathcal{X}}(x')$, and $\Omega_{\mathcal{Y}}$ defined analogously.

We formally state EGT [24] as Algorithm 1. The EGT algorithm alternates between taking steps focused on $\mathcal{X}$ and $\mathcal{Y}$. Algorithm 2 shows a single step focused on $\mathcal{X}$. Steps focused on $y$ are analogous. Algorithm 1 shows how the alternating steps and stepsizes are computed, as well as how initial points are selected.

Suppose the initial values $\mu_x, \mu_y$ satisfy $\mu_x = \frac{\varphi_{\mathcal{X}}}{L_1(f_{\mu_y})}$. Then, at every iteration $t \geq 1$ of EGT, the corresponding solution $z^t = [x^t; y^t]$ satisfies $x^t \in \mathcal{X}$, $y^t \in \mathcal{Y}$, the excessive gap condition is maintained, and

$$\epsilon_{\text{sad}}(x^T, y^T) \leq \frac{4\|A\|}{T+1} \sqrt{\frac{\Omega_{\mathcal{X}} \Omega_{\mathcal{Y}}}{\varphi_{\mathcal{X}} \varphi_{\mathcal{Y}}}}.$$

Consequently, EGT has a convergence rate of $O(\frac{1}{T})$ [24].

**Algorithm 1** EGT(DGF-center $x_\omega$, DGF weights $\mu_x, \mu_y$, and $\epsilon > 0$)

1: $x^0 = \nabla d_{\mathcal{X}}^* \left( \mu_x^{-1} \nabla f_{\mu_y}(x_\omega) \right)$
2: $y^0 = y_{\mu_y}(x_\omega)$
3: $t = 0$
4: **while** $\epsilon_{\text{sad}}(x^t, y^t) > \epsilon$ **do**
5: $\quad \tau_t = \frac{2}{t+3}$
6: $\quad$ **if** $t$ is even **then**
7: $\quad\quad (\mu_x^{t+1}, x^{t+1}, y^{t+1}) = \text{STEP}(\mu_x^t, \mu_y^t, x^t, y^t, \tau)$
8: $\quad$ **else**
9: $\quad\quad (\mu_y^{t+1}, y^{t+1}, x^{t+1}) = \text{STEP}(\mu_y^t, \mu_x^t, y^t, x^t, \tau)$
10: $\quad t = t + 1$
11: **return** $x^t, y^t$

**Algorithm 2** STEP$(\mu_x, \mu_y, x, y, \tau)$

1: $\hat{x} = (1 - \tau) x + \tau x_{\mu_x}(y)$
2: $y_+ = (1 - \tau) y + \tau y_{\mu_y}(\hat{x})$
3: $\tilde{x} = \nabla d_{\mathcal{X}}^* \left( \nabla d_{\mathcal{X}}(x_{\mu_x}(y)) - \frac{\tau}{(1-\tau)\mu_x} \nabla f_{\mu_y}(\hat{x}) \right)$
4: $x_+ = (1 - \tau) x + \tau \tilde{x}$
5: $\mu_x^+ = (1 - \tau) \mu_x$
6: **return** $\mu_x^+, x_+, y_+$

## 3 Treeplexes

Hoda et al. [11] introduced the *treeplex*, a class of convex polytopes that captures the sequence-form of the strategy spaces in perfect-recall EFGs.

**Definition 2.** *Treeplexes are defined recursively:*

1. Basic sets*: The standard simplex $\Delta_m$ is a treeplex.*

2. Cartesian product*: If $Q_1, \ldots, Q_k$ are treeplexes, then $Q_1 \times \cdots \times Q_k$ is a treeplex.*

3. Branching*: Given a treeplex $P \subseteq [0,1]^p$, a collection of treeplexes $Q = \{Q_1, \ldots, Q_k\}$ where $Q_j \subseteq [0,1]^{n_j}$, and $l = \{l_1, \ldots, l_k\} \subseteq \{1, \ldots, p\}$, the set defined by*

$$P \boxed{l} Q := \left\{ (x, y_1, \ldots, y_k) \in \mathbb{R}^{p + \sum_j n_j} : x \in P, \, y_1 \in x_{l_1} \cdot Q_1, \, \ldots, y_k \in x_{l_k} \cdot Q_k \right\}$$

*is a treeplex. We say $x_{l_j}$ is the branching variable for the treeplex $Q_j$.*

One interpretation of the treeplex is as a set of simplexes, where each simplex is weighted by the value of the variable above it in the parent branching operation (or 1 if there is no branching operation preceding the simplex). Thus the simplexes generally sum to the value of the parent rather than 1.

For a treeplex $Q$, we denote by $S_Q$ the index set of the set of simplexes contained in $Q$ (in an EFG $S_Q$ is the set of information sets belonging to the player). For each $j \in S_Q$, the treeplex rooted at the $j$-th simplex $\Delta^j$ is referred to as $Q_j$. Given vector $q \in Q$ and simplex $\Delta^j$, we let $\mathbb{I}_j$ denote the set of indices of $q$ that correspond to the variables in $\Delta^j$ and define $q^j$ to be the subvector of $q$ corresponding to the variables in $\mathbb{I}_j$. For each simplex $\Delta^j$ and branch $i \in \mathbb{I}_j$, the set $\mathcal{D}_j^i$ represents the set of indices of simplexes reached immediately after $\Delta^j$ by taking branch $i$ (in an EFG, $\mathcal{D}_j^i$ is the set of potential next-step information sets for the player). Given a vector $q \in Q$, simplex $\Delta^j$, and index $i \in \mathbb{I}_j$, each child simplex $\Delta^k$ for every $k \in \mathcal{D}_j^i$ is scaled by $q_i$. For a given simplex $\Delta^j$, we let $p_j$ denote the index in $q$ of the parent branching variable $q_{p_j}$ scaling $\Delta^j$. We use the convention that $q_{p_j} = 1$ if $Q$ is such that no branching operation precedes $\Delta^j$. For each $j \in S_Q$, $d_j$ is the maximum depth of the treeplex rooted at $\Delta^j$, that is, the maximum number of simplexes reachable through a series of branching operations at $\Delta^j$. Then $d_Q$ gives the depth of $Q$. We use $b_Q^j$ to identify the number of branching operations preceding the $j$-th simplex in $Q$. We say that a simplex $j$ such that $b_Q^j = 0$ is a *root simplex*.

Figure 1 illustrates an example treeplex $Q$. This treeplex $Q$ is constructed from nine two-to-three-dimensional simplexes $\Delta^1, \ldots, \Delta^9$. At level 1, we have two root simplexes, $\Delta^1, \Delta^2$, obtained by a Cartesian product operation (denoted by $\times$). We have maximum depths $d_1 = 2$, $d_2 = 1$ beneath them. Since there are no preceding branching operations, the parent variables for these simplexes $\Delta^1$ and $\Delta^2$ are $q_{p_1} = q_{p_2} = 1$. For $\Delta^1$, the corresponding set of indices in the vector $q$ is $\mathbb{I}_1 = \{1, 2\}$, while for $\Delta^2$ we have $\mathbb{I}_2 = \{3, 4, 5\}$. At level 2, we have the simplexes $\Delta^3, \ldots, \Delta^7$. The parent variable of $\Delta^3$ is $q_{p_3} = q_1$; therefore, $\Delta^3$ is scaled by the parent variable $q_{p_3}$. Similarly, each of the simplexes

$\Delta^3, \ldots, \Delta^7$ is scaled by their parent variables $q_{p_j}$ that the branching operation was performed on. So on for $\Delta^8$ and $\Delta^9$ as well. The number of branching operations required to reach simplexes $\Delta^1, \Delta^3$ and $\Delta^8$ is $b_Q^1 = 0, b_Q^3 = 1$ and $b_Q^8 = 2$, respectively.

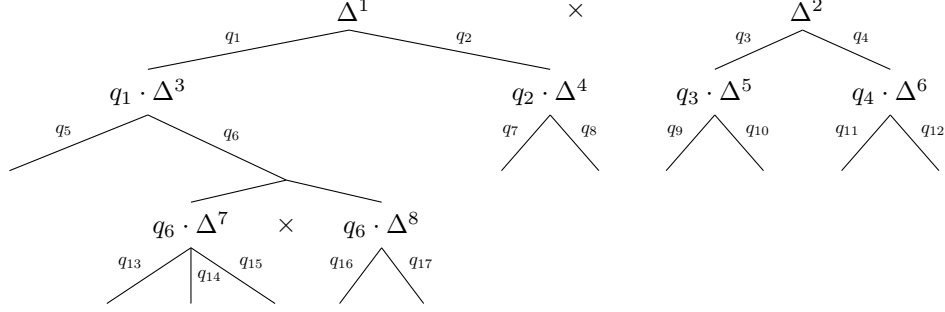

Figure 1: An example treeplex constructed from 9 simplexes. Cartesian product operation is denoted by $\times$.

## 4 Smoothed Best Responses

Let $d_j(x) = \sum_{i \in \mathbb{I}_j} x_i \log x_i + \log n$ be the entropy DGF for the $n$-dimensional simplex $\Delta_n$, where $n$ is the dimension of the $j$'th simplex in $Q$. Kroer et al. [19] introduced the following DGF for $Q$ by *dilating* $d_s$ for each simplex in $S_Q$ and take their sum: $d(q) = \sum_{j \in S_Q} \beta_j q_{p_j} d_j \left( \frac{q^j}{q_{p_j}} \right)$, where $\beta_j = 2 + \sum_{k \in \mathcal{D}^j} 2\beta_k$. Other dilated DGFs for treeplexes were introduced by Hoda et al. [11] and were also studied by Kroer et al. [17]. Kroer et al. [19] proved that this DGF is strongly convex modulus $\frac{1}{M}$ where $M$ is the maximum value of the $\ell_1$ norm over $Q$. EGT instantiated with this DGF converges at a rate of $\frac{LM^2 2^d \log n}{T}$ where $L$ is the maximum entry in the payoff matrix, $d$ is the depth of the treeplex, and $n$ is the maximum dimension of any individual simplex.

We now show how to solve (3) for this particular DGF. While it is known that this DGF has a closed-form solution, this is the first time the approach has been shown in a paper. Furthermore, we believe that our particular solution is novel, and leads to better control over numerical issues. The problem we wish to solve is the following.

$$\operatorname*{argmin} \sum_{j \in S_Q} \langle q^j, g_j \rangle + \beta_j q_{p_j} d_j(q^j / q_{p_j}) = \operatorname*{argmin} \sum_{j \in S_Q} q_{p_j} (\langle \bar{q}^j, g_j \rangle + \beta_j d_j(\bar{q}^j)) \qquad (4)$$

where the equality follows by the fact that $q_i = q_{p_j} \bar{q}_i$. For a leaf simplex $j$, its corresponding term in the summation has no dependence on any other part of the game tree except for the multiplication by $x_{p_j}$ (because none of its variables are parent to any other simplex). Because of this lack of dependence, the expression

$$\langle \bar{q}^j / q_{p_j}, g_j \rangle + \beta_j d_j(q^j / q_{p_j})$$

can be minimized independently as if it were an optimization problem over a simplex with variables $\bar{q}^j = x^j / q_{p_j}$ (this was also pointed out in Proposition 3.4 in Hoda et al. [11]). We show how to solve the optimization problem at a leaf: $\min_{\bar{q}^j \in \Delta_j} \langle \bar{q}^j, g_j \rangle + \beta_j d_j(\bar{q}^j)$. Writing the Lagrangian with respect to the simplex constraint and taking the derivative wrt. $\bar{q}_i$ gives

$$\min_{\bar{q}^j} \langle \bar{q}^j, g_j \rangle + \beta_j d_j(\bar{q}^j) + \lambda(1 - \sum_{i \in \mathbb{I}_j} \bar{q}_i) \Rightarrow g_i + \beta_j(1 + \log \bar{q}_i) = \lambda \Rightarrow \bar{q}_i \propto e^{-g_i/\beta_j}$$

This shows how to solve the smoothed-best-response problem at a leaf. For an internal simplex $j$, Proposition 3.4 of Hoda et al. [11] says that we can simply compute the value at all simplexes below $j$, add the value to $g_j$ (this is easily seen from (4); each $q_i$ acts as a scalar on the value of all simplexes

after $i$), and proceed by induction. Letting $|\mathbb{I}_j| = n$, we now simplify the objective function:

$$\langle \bar{q}^j, g_j \rangle + \beta_j (\sum_{i \in \mathbb{I}_j} (\bar{q}_i \log \bar{q}_i) + \log n) = \sum_i (\bar{q}_i(g_i + \beta_j \log \bar{q}_i)) + \beta_j \log n$$

$$= \sum_i (\bar{q}_i(\lambda - \beta_j)) + \beta_j \log n = \lambda - \beta_j + \beta_j \log n,$$

where the last two equalities follow first by applying our derivation for $\lambda$ and then the fact that $\bar{q}^j$ sums to one. This shows that we can choose an arbitrary index $i \in \mathbb{I}_j$ and propagate the value $g_i + \beta_j \log \bar{q}_i + \beta_j \log n$. In particular, for numerical reasons we choose the one that maximizes $\bar{q}_i$.

In addition to smoothed best responses, fast FOMs usually also require computation of proximal mappings, which are solutions to $\mathrm{argmin}_{q \in Q} \langle q, g \rangle + D(q \| q')$, where $D(q \| q') = d(q) - d(q') - \langle \nabla d(q'), q - q' \rangle$ is the Bregman divergence associated with the chosen DGF $d$. Unlike the smoothed best response, we are usually only interested in the minimizing solution and not the associated value. Therefore we can drop terms that do not depend on $q$ and the problem reduces to $\mathrm{argmin}_{q \in Q} \langle q, g \rangle + d(q) - \langle \nabla d(q'), q \rangle$, which can be solved with our smoothed best response approach by using the shifted gradient $\tilde{g} = g - \nabla d(q')$. This has one potential numerical pitfall: the DGF-gradient $\nabla d(q')$ may be unstable near the boundary of $Q$, for example because the entropy DGF-gradient requires taking logarithms. It is possible to derive a separate expression for the proximal mapping that is similar to what we did for the smoothed best response; this expression can help avoid this issue. However, because we only care about getting the optimal solution, not the value associated with it, this is not necessary. The large gradients near the boundary only affect the solution by setting bad actions too close to zero, which does not seem to affect performance.

## 5    Practical EGT

Rather than the overly conservative stepsize and $\mu$ parameters suggested in the theory for EGT we use more practical variants combining practical techniques from Kroer et al. [19] and Hoda et al. [11]. The pseudocode is shown in Algorithm 3. As in Kroer et al. [19] we use a practically-tuned initial choice for the initial smoothing parameters $\mu$. Furthermore, rather than alternating the steps on players 1 and 2, we always call STEP on the player with a higher $\mu$ value (this choice is somewhat reminiscent of the $\mu$-balancing heuristic employed by Hoda et al. [11] although our approach avoids an additional fitting step). The EGT algorithm with a practically-tuned $\mu$ and this $\mu$ balancing heuristic will be denoted EGT in our experiments. In addition, we use an EGT variant that employs the *aggressive $\mu$ reduction* technique introduced by Hoda et al. [11]. Aggressive $\mu$ reduction uses the observation that the original EGT stepsize choices, which are $\tau = \frac{2}{3+t}$, are chosen to guarantee the excessive gap condition, but may be overly conservative. Instead, aggressive $\mu$ reduction simply maintains some current $\tau$, initially set to $0.5$, and tries to apply the same stepsize $\tau$ repeatedly. After every step, we check that the excessive gap condition still holds; if it does not hold then we backtrack, $\tau$ is decreased, and we repeat the process. A $\tau$ that maintains the condition is always guaranteed to exist by Theorem 2 of Nesterov [24]. The pseudocode for this is given in Algorithm 4. EGT with aggressive $\mu$ reduction, a practically tuned initial $\mu$, and $\mu$ balancing, will be denoted EGT/AS in our experiments.

## 6    Algorithm Implementation

To compute smoothed best responses, we use a parallelization scheme. We parallelize across the initial Cartesian product of treeplexes at the root. As long as this Cartesian product is wide enough, the smoothed best response computation will take full advantage of parallelization. This is a common structure in real-world problems, for example representing the starting hand in poker, or some stochastic private state of each player in other applications. This parallelization scheme also works for gradient computation based on tree traversal. However, in this paper we do gradient computation by writing down a sparse payoff matrix using CUDA's sparse library and let CUDA parallelize the gradient computation.

For poker-specific applications (and certain other games where utilities decompose nicely based on private information) it is possible to speed up the gradient computation substantially by employing the

**Algorithm 3** EGT/AS(DGF-center $x_\omega$, DGF weights $\mu_x, \mu_y$, and $\epsilon > 0$)

1: $x^0 = \nabla d^*_{\mathcal{X}}\left(\mu_x^{-1}\nabla f_{\mu_y}(x_\omega)\right)$
2: $y^0 = y_{\mu_y}(x_\omega)$
3: $t = 0$
4: $\tau = \frac{1}{2}$
5: **while** $\epsilon_{\text{sad}}(x^t, y^t) > \epsilon$ **do**
6:    **if** $\mu_x > \mu_y$ **then**
7:       $(\mu_x^{t+1}, x^{t+1}, y^{t+1}, \tau) = \text{DECR}(\mu_x^t, \mu_y^t, x^t, y^t, \tau)$
8:    **else**
9:       $(\mu_y^{t+1}, y^{t+1}, x^{t+1}, \tau) = \text{DECR}(\mu_y^t, \mu_x^t, y^t, x^t, \tau)$
10:    $t = t + 1$
11: **return** $x^t, y^t$

**Algorithm 4** $\text{DECR}(\mu_x, \mu_y, x, y, \tau)$

1: $(\mu_x^+, x^+, y^+) = \text{STEP}(\mu_x, \mu_y, x, y, \tau)$
2: **while** $\text{EGV}(x, y) < 0$ **do**
3:    $\tau = \frac{1}{2}\tau$
4:    $(\mu_x^+, x^+, y^+) = \text{STEP}(\mu_x, \mu_y, x, y, \tau)$
5: **return** $\mu_x^+ x^t, y^t, \tau$

accelerated tree traversal of Johanson et al. [13]. We did not use this technique. In our experiments, the majority of time is spent in gradient computation, so this acceleration is likely to affect all the tested algorithms equally. Furthermore, since the technique is specific to games with certain structures, our experiments give a better estimate of general EFG-solving performance.

# 7 Experiments

We now present experimental results on running all the previously described algorithms on a GPU. All experiments were run on a Google Cloud instance with an NVIDIA Tesla K80 GPU with 12GB available. All code was implemented in C++ using CUDA for GPU operations, and cuSPARSE for the sparse payoff matrix. We compare against several CFR variants.[1] We run CFR with RM (CFR(RM)), RM$^+$ (CFR(RM$^+$)), and CFR$^+$ which is CFR with RM$^+$ and a linear averaging scheme. We now describe these variants. Detailed descriptions can also be found in Zinkevich et al. [29] and Tammelin et al. [27].

Our experiments are conducted on real large-scale "river" endgames faced by the *Libratus* AI [4]. *Libratus* was created for the game of heads-up no-limit Texas hold'em. *Libratus* was constructed by first computing a "blueprint" strategy for the whole game (based on abstraction and Monte-Carlo CFR [20]). Then, during play, *Libratus* would solve endgames that are reached using a significantly finer-grained abstraction. In particular, those endgames have no card abstraction, and they have a fine-grained betting abstraction. For the beginning of the subgame, the blueprint strategy gives a conditional distribution over hands for each player. The subgame is constructed by having a Chance node deal out hands according to this conditional distribution.[2]

A subgame is structured and parameterized as follows. The game is parameterized by the conditional distribution over hands for each player, current pot size, board state (5 cards dealt to the board), and a betting abstraction. First, Chance deals out hands to the two players according to the conditional hand distribution. Then, Libratus has the choice of folding, checking, or betting by a number of multipliers of the pot size: 0.25x, 0.5x, 1x, 2x, 4x, 8x, and all-in. If Libratus checks and the other player bets then Libratus has the choice of folding, calling (i.e. matching the bet and ending the betting), or raising by pot multipliers 0.4x, 0.7x, 1.1x, 2x, and all-in. If Libratus bets and the other player raises Libratus can fold, call, or raise by 0.4x, 0.7x, 2x, and all-in. Finally when facing subsequent raises Libratus can fold, call, or raise by 0.7x and all-in. When faced with an initial check, the opponent can fold, check, or raise by 0.5x, 0.75x, 1x, and all-in. When faced with an initial bet the opponent can fold, call, or raise by 0.7x, 1.1x, and all-in. When faced with subsequent raises the opponent can fold, call, or raise by 0.7x and all-in. The game ends whenever a player folds (the other player wins all money in the pot), calls (a showdown occurs), or both players check as their first action of the game (a showdown occurs). In a showdown the player with the better hands wins the pot. The pot is split in case of a tie. (For our experiments we used endgames where it is *Libratus*'s turn to move first.)

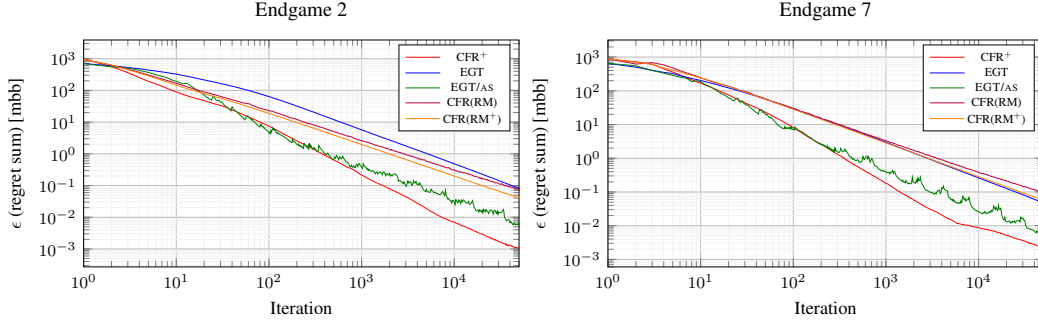

Figure 2: Solution quality as a function of the number of iterations for all algorithms on two river subgames. The solution quality is given as the sum of regrets for the players in milli-big-blinds.

We conducted experiments on two river endgames extracted from *Libratus* play: Endgame 2 and Endgame 7. Endgame 2 has a pot of size 2100 at the beginning of the river endgame. It has dimension 140k and 144k for *Libratus* and the opponent, respectively, and 176M leaves in the games tree. Endgame 7 has a pot of size $3750 at the beginning of the river subgame. It has dimension 43k and 86k for the players, and 54M leaves.

In the first set of experiments we look at the per-iteration performance of each algorithm. The results are shown in Figure 2. The y-axis shows the sum of the regrets for each player, that is, how much utility they can gain by playing a best response rather than their current strategy. The unit is milli-big-blinds (mbb); at the beginning of the original poker game, *Libratus*, as the "big blind", put in $100 and the opponent put in $50, in order to induce betting. Mbb is a thousandth of the big blind value, that is, 10 cents. This is a standard unit used in research that uses poker games for evaluation. One mbb is often considered the convergence goal. CFR$^+$ and EGT/AS perform the best; both reach the goal of 1mbb after about 400 iterations in both Endgame 2 and 7. EGT, CFR(RM), and CFR(RM$^+$) all take about 3000 iterations to reach 1mbb in Endgame 7. In Endgame 2, EGT is slowest, although the slope is steeper than for CFR(RM) and CFR(RM$^+$). We suspect that better initialization of EGT could lead to it beating both algorithms. Note also that EGT was shown better than CFR(RM) and CFR(RM$^+$) by Kroer et al. [19] in the smaller game of Leduc hold'em with an automated $\mu$-tuning approach. Their results further suggest that better initialization may help enhance converge speed significantly.

One issue with per-iteration convergence rates is that the algorithms do not perform the same amount of work per iteration. All CFR variants in our experiments compute 2 gradients per iteration, whereas EGT computes 3, and EGT/AS computes 4 (the additional gradient computation is needed in order to evaluate the excessive gap). Furthermore, EGT/AS may use additional gradient computations if the excessive gap check fails and a smaller $\tau$ is tried (in our experiments about 15 adjustments were needed). In our second set of plots, we show the convergence rate as a function of the total number of gradient computations performed by the algorithm. This is shown in Figure 3. By this measure, EGT/AS and EGT perform slightly worse relative to their performance as measured by iteration count. In particular, CFR$^+$ takes about 800 gradient computations in order to reach 1mbb in either game, whereas EGT/AS takes about 1800.

In our experiments CFR$^+$ vastly outperforms its theoretical convergence rate (in fact, every CFR variant does significantly better than the theory predicts, but CFR$^+$ especially so). However, CFR$^+$ is known to eventually reach a point where it slows down and performs worse than $\frac{1}{T}$. In our experiments we start to see CFR$^+$ slowing down towards the end of Endgame 7. EGT, in contrast, is guaranteed to maintain a rate of $\frac{1}{T}$, and so may be preferable if a guarantee against slowdown is desired or high precision is needed.

## 8   Conclusions and Future Research

We introduced a practical variant of the EGT algorithm that employs aggressive stepsizes, $\mu$ balancing, a numerically-friendly smoothed-best-response algorithm, parallelization via Cartesian product operations at the root of the strategy treeplex, and a GPU implementation. We showed for the first time, via experiments on real large-scale *Libratus* endgames, that FOMs (with the dilated entropy

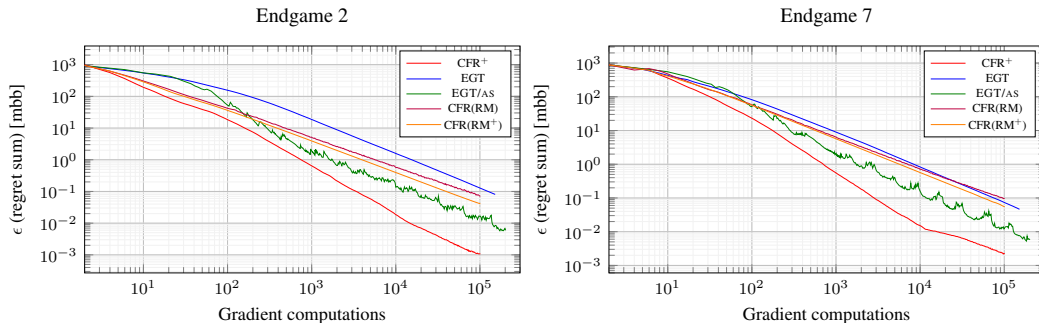

Figure 3: Solution quality as a function of the number of gradient computations for all algorithms on two river subgames. The solution quality is given as the sum of regrets for the players in milli-big-blinds.

DGF) are competitive with the CFR family of algorithms. Specifically, they outperform the other CFR variants and are close in efficiency to CFR$^+$. Our best variant of EGT can solve subgames to the desired accuracy at a speed that is within a factor of two of CFR$^+$.

Our results suggest that it may be possible to make FOMs faster than CFR$^+$. For example, we did not spend much effort tuning the parameters of EGT, and tuning them would make the algorithm even more efficient. Second, we only investigated EGT, which has been most popular FOM in EFG solving. However, it is possible that other FOMs such as mirror prox [23] or the primal-dual algorithm by Chambolle and Pock [8] could be made even faster.

Furthermore, stochastic FOMs (i.e., ones where the gradient is approximated by sampling to make the gradient computation dramatically faster) could be investigated as well. Kroer et al. [17] tried this using *stochastic mirror prox* [14] without practical success, but it is likely that this approach could be made better with more engineering.

It would also be interesting to compare our EGT approach to CFR algorithms for computing equilibrium refinements, for example in the approximate extensive-form perfect equilibrium model investigated by Kroer et al. [18] and Farina et al. [9].

Pruning techniques (for temporarily skipping parts of the game tree on some iterations) have been shown effective for both CFR and EGT-like algorithms, and could potentially be incorporated as well [20, 6, 2].

Finally, while EGT, as well as other FOM-based approaches to computing zero-sum Nash equilibria, are not applicable to the computation of general-sum Nash equilibria in theory they could still be applied to the computation of strategies in practice (gradients can still be computed, and so the smoothed best responses and corresponding strategy updates are still well-defined). For CFR the analogous approach seems to perform reasonably well [7], and you might expect the same from FOMs such as EGT.

**Acknowledgments** This material is based on work supported by the National Science Foundation under grants IIS-1718457, IIS-1617590, and CCF-1733556, and the ARO under award W911NF-17-1-0082. Christian Kroer is supported by a Facebook Fellowship.

## Footnotes

[1] All variants use the alternating updates scheme.

[2] *Libratus* used two different subgame-solving techniques, one "unsafe" and one "safe" [3]. The computational problem in the two is essentially identical. We experiment with the "unsafe" version, which uses the prior distributions described here.

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
