[Supplementary Material · gpu_egt_river.nips2018.supplementary.pdf]

# A  Prox shift

We want to find a simple, numerically-friendly expression for

$$-d(q) + \langle \nabla d(q), q \rangle$$

First we derive an expression for $\langle \nabla d(q), q \rangle$. Let $i \in \mathbb{I}_j$ and $n_j$ be the dimensionality of simplex $j$. Taking derivatives we have

$$\nabla_{ji} d(q) = \beta_j (\log \frac{q_i}{q_{p_j}} + 1) + \sum_{k \in \mathcal{D}_j^i} \beta_k \left( \log n_k - \sum_{i' \in \mathbb{I}_k} \frac{x_{i'}}{x_{p_k}} \right)$$

$$= \beta_j (\log \frac{q_i}{q_{p_j}} + 1) + \sum_{k \in \mathcal{D}_j^i} \beta_k (\log n_k - 1)$$

Taking the inner product with $q$ gives

$$\langle \nabla d(q), q \rangle = \sum_{j \in S_Q; i \in \mathbb{I}_j} q_i \left[ \beta_j (\log \frac{q_i}{q_{p_j}} + 1) + \sum_{k \in \mathcal{D}_j^i} \beta_k (\log n_k - 1) \right]$$

Subtracting $-d(q)$ gives

$$-d(q) + \langle \nabla d(q), q \rangle = - \sum_{j \in S_Q; i \in \mathbb{I}_j} \beta_j q_i \log \frac{q_i}{q_{p_j}} - \sum_{j \in S_Q} \beta_j q_{p_j} \log(n_j)$$

$$+ \sum_{j \in S_Q; i \in \mathbb{I}_j} q_i \left[ \beta_j (\log \frac{q_i}{q_{p_j}} + 1) + \sum_{k \in \mathcal{D}_j^i} \beta_k (\log n_k - 1) \right]$$

$$= \sum_{j \in S_Q; i \in \mathbb{I}_j} q_i \beta_j + \sum_{j \in S_Q} \sum_{k \in \mathcal{D}_j^i} q_{p_k} \beta_k (\log n_k - 1) - \sum_{j \in S_Q} \beta_j q_{p_j} \log(n_j)$$

$$= \sum_{j \in S_Q} q_{p_j} \beta_j + \sum_{j \in S_Q} \sum_{k \in \mathcal{D}_j^i} q_{p_k} \beta_k (\log n_k - 1) - \sum_{j \in S_Q} \beta_j q_{p_j} \log(n_j)$$

$$= \sum_{j \in S_Q} \sum_{k \in \mathcal{D}_j^i} q_{p_k} \beta_k (\log n_k - 1) - \sum_{j \in S_Q} \beta_j q_{p_j} (\log(n_j) - 1)$$

$$= - \sum_{j \in S_Q; b_Q^j = 0} \beta_j q_{p_j} (\log(n_j) - 1)$$