[Reviews · NeurIPS 2018]

Reviewer 1



Previous results on zero sum imperfect information games such as poker have mostly used counterfactual regret minimization (CFR) and variants to find approximate equilibria, even though CFR has worse theoretical guarantees than first order methods (FOM). This paper shows that a specific first order method called the excessive gap technique with aggressive step size (EGT/AS) is competitive with CFR and it's variants. This means that it performs worse than the best CFR variants, but better than others. The paper is a somewhat incremental extension of previous work laying out the theory of EGT and the specific distance functions needed. Previous work also introduced EGT/AS, and showed that it worked on small to medium sized games. The main contributions of this paper are 1. Showing that EGT/AS works reasonably well on large games, specifically poker endgames arising from Libratus. 2. Some technical and numerical details concerning solving EGT subprograms in numerically stable ways, and a bit of (relatively straightforward) discussion about mapping these computations to GPUs. I would advocate accepting this paper, as demonstrating performance of EGT/AS on interesting games is a useful contribution, and is useful for guiding future research. The sense I get from this paper is that the space of algorithms spanned by CFR, EGT, and their variants is far from settled, and it seems likely that future work will find better variants that match the best theoretical guarantees with better empirical performance. That blending isn't this paper, but it's useful to know that EGT/AS is an empirical contender (even if it isn't as good as the best CFR variants for reasonable amounts of convergence). As a general aside: I am curious if or when when poker algorithms will adopt actual ML techniques. It's reasonable to include this series of "clever brute force, non-ML" algorithms in NIPS since they are solving a goal clearly within the scope of NIPS applications, but it's interesting that poker seems to be going the way of chess rather than go and avoi ding primarily ML solutions. On the subject of reproducibility: the methods presented are clear, so the main reproducibility barrier is having the surrounding machinery for a Libratus-equivalent engine. This is hard to avoid for this type of scaling paper, so it should not count much against acceptance. Various notes: 1. The abstract should say "imperfect information" clearly. It's mathematically implied by not saying "perfect information", but worth being explicit. 2. The first sentence frightens me a little bit (line 21-22). Hopefully most business and military games are not zero sum. :) 3. Line 99: "and is strongly convex modulus" should be "and has...". 4. Line 108: What norm is ||A||? From Nesterov it looks like a mixed p-norm operator norm, which no one is going to guess without looking at Nesterov, and I'm confused by the norm would be asymmetrically defined given the symmetric structure of the bilinear program. Also, what are the a_1, a_2 constants? Ah, are those the unnecessary linear terms in the bilinear function phi? Finally, it's bad to introduce phi in line 85, discard it, and then introduce another function named phi in equation (3). 6. Why is the (E)xcessive (G)ap (C)ondition called EGV? 7. It's worth some discussion whether EGT/AS or variants are applicable to the nonzero sum setting, and more generally to games with more than 2 players. By default I believe the answer is no, since two player zero sum has additionally convexity structure which these algorithms are exploiting heavily.

Reviewer 2



The key contribution of this paper is empirical evaluation of a new, numerical friendly GPU implementation of the EGT method for solving extensive form games. The evaluation is done against state of the art CFR/CFR+ variants. Quality: + The overall quality of the paper is high + The evaluation is done on a meaningful games + Authors tried hard to make the comparison fair, and I believe they did a good job in doing so. - While authors mention the GPU implementation multiple times, there’s no analysis nor discussion what speedups this actually brings. Since the results (Figure 1) are all compared in the number of iterations, it should look exactly the same if the implementation would be on CPU. -The experimental section is near impossible to reproduce. The reason is that the games the authors run the experiments on are the subgames reached by Libratus agent. These are not accessible which means that a) no-one can really reproduce the graphs b) somewhat breaks blind submission since it’s clear who has access to these subgames. Clarity: + The paper is well written and structured Originality: + Authors do a good job in explaining the previous work - The actual contribution from prior work is not large, the paper is mostly about numerical stable implementation on GPUs and evaluation. Significance: + The results are for sure interesting for the community - The results are not particularly surprising, pretty much align with what previous research would suggest

Reviewer 3



The paper details a first-order method (FOM) for solving extensive-form games, based on Nesterov's Excessive Gap Technique, and evaluates it against methods based on counterfactual regret minimization (CFR) on the problem of solving endgames in no-limit Texas hold'em. My high overall score is justified by my belief in the importance of the problem domain (i.e., solving large extensive-form games) and the apparent novelty and significance of the "take-away message" that first-order methods can be nearly as effective as counterfactual regret minimization for these problems. My low confidence score is based on the fact that I have no experience working in this research area and have not previously read papers on solving large extensive-form games, so I have no basis for judging this paper in relation to existing research in the area. QUALITY: My remarks below about clarity, originality, and significance completely summarize my view of the paper's quality. CLARITY: In general, I appreciated the clarity of the exposition of background information in Sections 1, 2, and 3, and the clarity of the sections describing the evaluation (Sections 6 and 7). Specific comments and requests for clarification are included below. ORIGINALITY: The paper relies heavily on prior work of Hoda et al. and Kroer et al. The originality of the present work appears to lie in the specific combination of techniques adopted from those prior works, and (more importantly) in the implementation and evaluation reported in Sections 6 and 7. SIGNIFICANCE: First-order methods represent quite a different approach to solving bilinear saddle point problems, so it is noteworthy that these methods are nearly competitive with the best CFR-based solvers on a large-scale real-world application. Needless to say, I would deem the result to be even more significant if the paper had succeeded in developing a first-order method that was outright superior to CFR, but I feel the paper succeeds in showing that EGT holds enough promise that it merits closer investigation in future research. Minor comments: * page 3, line 109: what is Q? Does it refer to an arbitrary convex set? * page 4, final paragraph of section 3: I found the paragraph confusing. It refers to the set of simplexes contained in Q, but according to the recursive definition of treeplexes, their constituent pieces are treeplexes but not necessarily simplexes. * page 5, lines 168-174: please explain what "propagate the value" means. * page 7, line 254: what does "sequences" mean in this context? UPDATE: The author feedback addressed all of my minor comments. Thank you for explaining these points!